# Characterization and Optimal Design of Silicon-Rich Nitride Nonlinear Waveguides for 2 μm Wavelength Band

**Zhihua Tu [1], Daru Chen [2], Hao Hu [3], Shiming Gao [1,4] and Xiaowei Guan [3,]** 

[1] Centre for Optical and Electromagnetic Research, State Key Laboratory of Modern Optical Instrumentation, International Research Centre for Advanced Photonics, Zhejiang University, Hangzhou 310058, China; tuzhihua@zju.edu.cn (Z.T.); gaosm@zju.edu.cn (S.G.)

[2] Hangzhou Institute of Advanced Studies, Zhejiang Normal University, Hangzhou 311231, China; daru@zjnu.cn

[3] DTU Fotonik, Department of Photonics Engineering, Technical University of Denmark, 2800 Kgs. Lyngby, Denmark; huhao@fotonik.dtu.dk

[4] Ningbo Research Institute, Zhejiang University, Ningbo 315100, China

[*] Correspondence: xgua@fotonik.dtu.dk

**Abstract:** Optical communication using the 2 μm wavelength band is attracting growing attention for the sake of mitigating the information 'capacity crunch' on the way, where on-chip nonlinear waveguides can play vital roles. Here, silicon-rich nitride (SRN) ridge waveguides with different widths and rib heights are fabricated and measured. Linear characterizations show a loss of ~2 dB/cm of the SRN ridge waveguides and four-wave mixing (FWM) experiments with a continuous wave (CW) pump reveal a nonlinear refractive index of ~$1.13 \times 10^{-18}$ m$^2$/W of the SRN material around the wavelength 1950 nm. With the extracted parameters, dimensions of the SRN ridge waveguides are optimally designed for improved nonlinear performances for the 2 μm band, i.e., a maximal nonlinear figure of merit (i.e., the ratio of nonlinearity to loss) of 0.0804 W$^{-1}$ or a super-broad FWM bandwidth of 518 nm. Our results and design method open up new possibilities for achieving high-performance on-chip nonlinear waveguides for long-wavelength optical communications.

**Keywords:** four-wave mixing; nonlinear figure of merit; silicon-rich nitride; ridge waveguide; conversion bandwidth

## 1. Introduction

The capacity of the current optical communication systems operating at the mainstream wavelengths, i.e., around 1310 nm and 1550 nm is approaching a shortage due to the explosive growth in information being transmitted. In order to exploit new frequency resources for telecommunications, transmitting signals at mid-infrared (MIR) wavelengths has been put on the agenda [1–6], among which the 2 μm wavelength band (1900 to 2100 nm) attracts special attention as signals in this band can be significantly amplified by thulium-doped fiber amplifiers (TDFAs) [7] and glass fiber in this band is still transparent [8]. Indeed, besides the amplifier, many other functional devices necessary for the 2-μm-band optical communication system have been also commercially available or laboratory developed including the laser [9,10], modulator [11], low-loss hollow-core fiber [12], wavelength-division multiplexer [13,14], detector [15,16], etc. Meanwhile, nonlinear devices like the wavelength converter are also indispensable and, in order to leverage the manufacturing scalability for such devices, it is preferred to use the on-chip nonlinear waveguides with the fabrication

processes compatible with the mature silicon-based complementary metal-oxide-semiconductor (CMOS) technology.

As to the CMOS-compatible nonlinear waveguides, silicon (Si) waveguides and stoichiometric silicon nitride ($Si_3N_4$) waveguides are currently the workhorses at the telecommunication wavelengths [17–21]. However, Si still suffers from the two-photon absorption (TPA) below the wavelength 2200 nm [22], despite that linear Si waveguides can possess a low loss in the 2 μm band [23] and the Si nonlinear waveguides have been used for parametric conversion around 2 μm [24]. Indeed, key nonlinear devices like the parametric amplifier [25] or the frequency comb generator [26] have been demonstrated using the Si nonlinear waveguides in the MIR wavelengths but just over the wavelength 2200 nm. For $Si_3N_4$, its low linear refractive index (RI) inevitably gives a large footprint of any $Si_3N_4$-based photonic device and the low nonlinear RI usually produces a poor energy efficiency of devices based on the $Si_3N_4$ nonlinear waveguides. Moreover, in spite of demonstrations of high-performance $Si_3N_4$ waveguides fabricated by using some advanced processes [27], strain issues are still present in $Si_3N_4$ film, especially in the thick $Si_3N_4$ film, which are yet necessary for achieving compact devices at long wavelengths like 2 μm and for waveguides with proper dispersion. Besides, titanium dioxide ($TiO_2$) waveguides [4] and silicon germanium (SiGe) waveguides [28] were also demonstrated to transmit optical signals at 2 μm, but, up to now, no reports can be found for them being used for nonlinear processes in this band.

Recently, silicon-rich nitride (SRN) has emerged as a promising candidate for the on-chip nonlinear optics [29–39] since SRN possesses larger linear and nonlinear RIs from a Si excess compared to the stoichiometric $Si_3N_4$ and, more importantly, the SRN film can have a substantially reduced stress [40]. While the SRN waveguides have been applied for many nonlinear processes like the octave-spanning supercontinuum generation (SCG) [31] and the frequency comb generation (FCG) [32], they were almost operating at the near infrared wavelengths. A SRN waveguide was very recently used to transmit optical signals at 2 μm and the wavelength conversion was also demonstrated with a pulse pump in the 2 μm band, but the waveguide was quite thin (300 nm) and the dispersion was not optimized [39]. Here, we design and fabricate SRN ridge waveguides and perform linear and nonlinear characterizations of them, which show a propagation loss of ~2 dB/cm and a moderate nonlinearity of ~4.62 $W^{-1}$ $m^{-1}$ of the fabricated SRN waveguides, corresponding to a nonlinear RI of ~$1.13 \times 10^{-18}$ $m^2$/W of the SRN material at the wavelength 1.95 μm. Then the SRN nonlinear ridge waveguides are optimized by adjusting the width and the rib height to achieve the maximal nonlinear figure of merit (FOM) and to achieve a broad four-wave mixing (FWM) bandwidth of 518 nm.

## 2. Structure and Fabrication

Figure 1a shows the schematic structure of the present SRN ridge waveguide with the total height, rib height and width denoted as *H*, *h*, and *w*, respectively. A thick SRN layer was needed for tightly confining light in the SRN core and obtaining a proper dispersion profile. Here, we used a low-pressure chemical vapor deposition (LPCVD) machine to deposit an 810 nm SRN film on 2.5 μm thermal silicon dioxide ($SiO_2$). The SRN film was first deposited by a reaction between dichlorsilane ($SiH_2Cl_2$ at 80 sccm) and ammonia ($NH_3$ at 20 sccm) at 830 °C and a pressure of 120 mTorr, and then annealed for three hours at 1150 °C. After annealing, the Si excess was measured to be ~18.7% and the SRN film was condensed to 800 nm. A thick resist (CSAR, ~1.1 μm) was then spun and patterned by the electron beam lithography. The patterns were then transferred to the SRN film by using an inductive coupled plasma machine at –10 °C with flows of the etching gases $SF_6$/Ar/$CF_4$/$CH_4$ being 4/10/4/2 sccm. SRN ridge waveguides with various widths and rib heights were fabricated with a fixed *H* of 800 nm. After residual resist stripping, the ridge waveguides were finalized with air as the upper cladding and the dimensions were measured, as shown by an exemplified scanning electron microscopy (SEM) image in Figure 1b. RIs at near infrared wavelengths of the deposited and annealed SRN film were measured by an ellipsometry method and fitted with the Sellmeier formula to extract the RIs around 2 μm wavelengths. Here, the RI was calculated to be 2.123 at 1950 nm. Figure 1c shows the simulated

mode profile of the fundamental transverse-electric (TE$_0$) mode for a SRN ridge waveguide with $w$ = 1.32 μm and $h$ = 640 nm at the wavelength 1950 nm. The power confinement ratio in the SRN core was calculated to be 89.2%, indicating strong confinement of the SRN ridge waveguide.

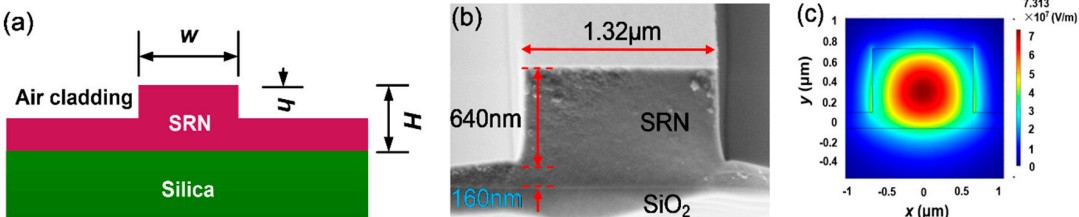

**Figure 1.** The fabricated silicon-rich nitride (SRN) ridge waveguide. (**a**) Schematic diagram of the structure. (**b**) Scanning electron microscopy image. (**c**) Simulated fundamental transverse-electric (TE$_0$) mode profile at 1950 nm of the waveguide with a width of 1.32 μm and a rib height of 640 nm.

## 3. Linear and Nonlinear Characterizations

### 3.1. Linear Characterizations

Linear transmission spectra of the fabricated SRN ridge waveguides with lengths of 5.5 mm, 13 mm and 21 mm were measured from 1860 nm to 1980 nm. Then the cut-back method was used to extract the propagation losses and the coupling losses to a tapered lensed fiber for waveguides with different $w$ and $h$. In order to reduce the accidental error, six identical waveguides were fabricated and measured for each waveguide dimension configuration. To serve as an example, Figure 2a,b exhibits the propagation and coupling losses with respect to the wavelength for the fabricated SRN ridge waveguides with $h$ = 380 nm and $w$ = 950 nm, and $h$ = 640 nm and $w$ = 1320 nm, respectively. Due to the Fabry–Perot interference effect between the two facets of a waveguide, the nonperfect uniformity of the six identical waveguides and the measurement inaccuracy, there were statistical errors for the extracted losses, as shown in the figures. Nevertheless, the propagation losses of the fabricated SRN ridge waveguides were around 2 dB/cm in the investigated wavelength range. Meanwhile, the coupling losses were around 4 dB/facet. Results on the other waveguide dimensions are summarized in Table 1 in the end of this section.

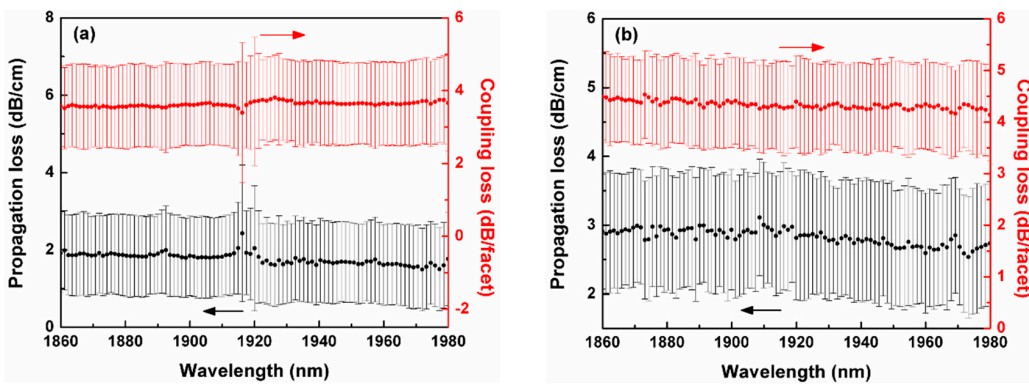

**Figure 2.** The measured propagation loss and coupling loss with respect to the wavelength for the fabricated silicon-rich nitride ridge waveguides with (**a**) $h$ = 380 nm and $w$ = 950 nm and (**b**) $h$ = 640 nm and $w$ = 1320.

The propagation loss $\alpha$ of the fabricated SRN ridge waveguide includes linear absorption loss $\alpha_1$ and scattering loss $\alpha_2$. $\alpha_1$ can be derived from the imaginary part of the effective RI, $N_{eff,r}$ of the SRN ridge waveguide by considering the absorption of the SRN material and expressed as [41]

$$\alpha_1 = \frac{2\omega}{c}\text{Im}(N_{eff,r}) \tag{1}$$

where $\omega$ is the optical angular frequency and $c$ is the speed of light in vacuum. The scattering loss $\alpha_2$ depends on the sidewall roughness, half the waveguide width $w/2$, RI steps from the SRN core to the surroundings, and scaling factor of a ridge waveguide to a channel waveguide (i.e., $h = H$). The sidewall roughness is characterized by the correlation length $L_c$ and the mean squared error $\sigma^2$ deviated from the sidewall surface. Thus, the scattering loss $\alpha_2$ can be expressed as Equation (2) [42]

$$\alpha_2 = 4.343 \frac{\sigma^2}{\sqrt{2}k_0(w/2)^4 N_{eff,r}} g \cdot f \cdot s \tag{2}$$

where the unit of $\alpha_2$ is dB/m, $k_0$ is the wave vector in vacuum, and function $g$ is completely determined by the dimensions of the SRN channel waveguide and $f$ is relevant to $L_c$ and RI steps. Details for calculating $g$ and $f$ can be found in Payne and Lacey's work for analyzing a planar waveguide's scattering loss using the exponential autocorrelation function [43]. The scaling factor $s$ can be calculated as [42]

$$s = \frac{\delta N_{eff,r}/\delta d}{\delta N_{eff,c}/\delta d} \tag{3}$$

where $N_{eff,c}$ is the effective RI of the SRN channel waveguide and $d$ is half the waveguide width $w/2$.

A larger $h$ and a smaller $w$ mean more SRN being etched and more light fields overlapping the rough sidewalls and thus a larger scattering loss. In contrast, for the SRN ridge waveguide with a small $h$ and a large $w$, almost all the light is confined inside the SRN core and, thus, $\alpha$ is dominantly determined by the absorption loss $\alpha_1$. Here, we reasonably assume $\alpha_1$ to be 1.7 dB/cm considering it is the minimal propagation loss value we have extracted for the waveguides with different dimensions and from the ridge waveguide with a very shallow etch, i.e., $h = 230$ nm and $w = 840$ nm. Thus, with Equation (2), we can use the measured propagation losses of the SRN ridge waveguides with different dimensions to fit $L_c$ and $\sigma^2$. The fitted values of (5, 11.3) nm for ($\sigma$, $L_c$) were found to be able to provide nice fittings of the calculated losses using Equation (2) to the measured propagation losses, as shown in Figure 3, and meanwhile consistent with the values ($\sigma = 5$ nm, $L_c = 45$ nm) reported in other literature where the sidewall roughness of a fabricated SRN waveguide was characterized [30]. Thus, it is reasonable to use Equation (2) to predict propagation losses of the SRN ridge waveguide with any $w$ and $h$.

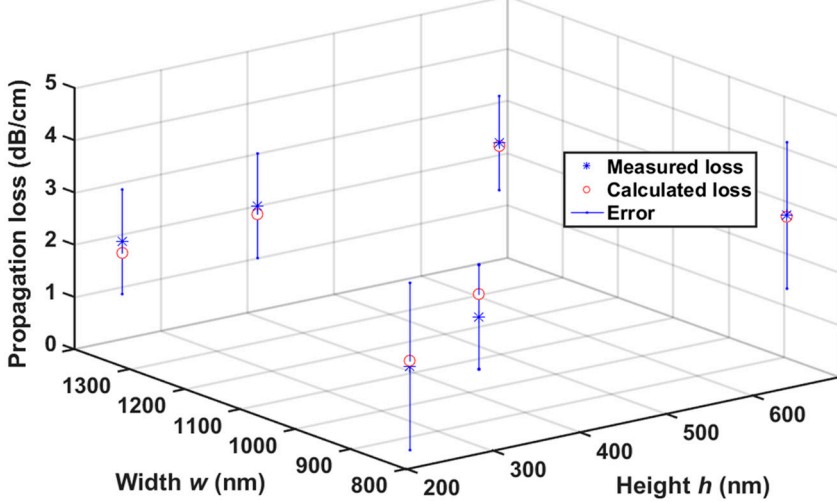

**Figure 3.** The measured and calculated propagation losses of the fabricated silicon-rich nitride ridge waveguides with different dimensions.

### 3.2. Nonlinear Characterizations

With the experimental setup shown in Figure 4, FWM experiments were implemented to characterize the nonlinear parameter $\gamma$ of the fabricated SRN ridge waveguides with different dimensions. The pump signal was provided by a commercial continuous-wave (CW) laser (AdValue Photonics AP-CW1) with a fixed wavelength of 1950.1 nm. The probe signal was generated by a homemade CW laser having a thulium doped fiber as the gain media pumped at 793 nm. The wavelength of our homemade CW laser was set at 1953 nm. High power isolators (HP-ISOs) were used to protect the CW lasers and the pump and signal lights were combined by a 9:1 coupler. The combination of a fiber polarization beam splitter (PBS) and a fiber polarization controller (PC) was implemented to guarantee both the pump light and the signal light were TE-polarized when injected into the waveguides by a tapered lensed fiber. The output light was collected by another lensed fiber and the FWM spectra were recorded by an optical spectrum analyzer (OSA, Yokogawa AQ6375B).

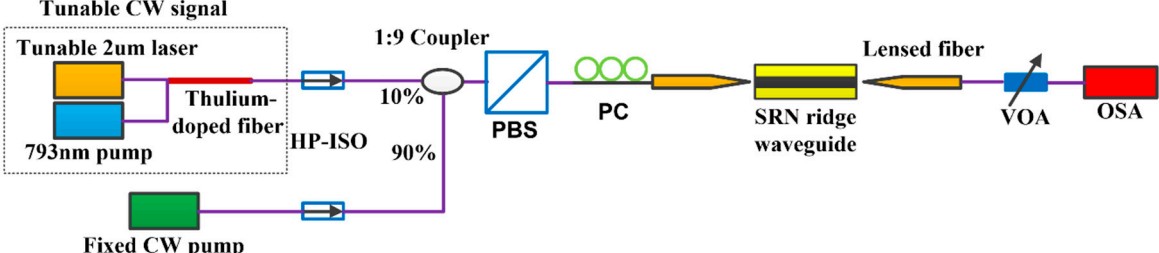

**Figure 4.** The experimental setup for four-wave mixing (FWM) experiments in the fabricated silicon-rich nitride (SRN) ridge waveguides. CW: continuous-wave; HP-ISO: high power isolator; PBS: polarization beam splitter; PC: polarization controller; VOA: variable optical attenuator; OSA: optical spectrum analyzer.

The idler signal with $\omega_I = 2\omega_P - \omega_S$ can be generated in the SRN ridge waveguide via the degenerate FWM, where $\omega_I$, $\omega_P$, $\omega_S$ are the frequencies of the idler, pump and signal lights, respectively. Since the deposited SRN possessing a moderate excess of Si, the TPA and free-carrier absorption (FCA) can be neglected in the 2 μm band. Considering the linear loss, self-phase modulation, cross-phase modulation and FWM, the interaction relationships among these three lights can be described by the following full-vectorial coupling equations [41]

$$\frac{\partial A_P}{\partial z} = -\frac{1}{2}\alpha_P A_P + j\gamma_P\left(|A_P|^2 + 2|A_S|^2 + 2|A_I|^2\right)A_P + 2j\gamma_P A_S A_I A_P^* \exp(j\Delta\beta z) \tag{4}$$

$$\frac{\partial A_S}{\partial z} = -\frac{1}{2}\alpha_S A_S + j\gamma_S\left(|A_S|^2 + 2|A_P|^2 + 2|A_I|^2\right)A_S + j\gamma_S A_P^2 A_I^* \exp(-j\Delta\beta z) \tag{5}$$

$$\frac{\partial A_I}{\partial z} = -\frac{1}{2}\alpha_I A_S + j\gamma_I\left(|A_I|^2 + 2|A_P|^2 + 2|A_S|^2\right)A_I + j\gamma_I A_P^2 A_S^* \exp(-j\Delta\beta z) \tag{6}$$

where $A_m$ ($m = P, S, I$) is the field amplitude of the pump, signal, or idler, $\alpha_m$ is the linear loss, $\Delta\beta$ is the linear phase mismatch, and $\gamma_m$ is the nonlinear parameter, respectively. When the interacting waves are all in the same wavelength band, $\gamma_m$ can be expressed as Equation (7) [44]

$$\gamma_m = \frac{\omega_m}{cA_{eff}(\omega_m)}\overline{n}_2(\omega_m) \tag{7}$$

where $A_{eff}$ is the effective mode area and $\bar{n}_2$ is the effective nonlinear RI. They can be calculated as [45]

$$A_{eff}(\omega_m) = \frac{\left| \iint \text{Re}\left[ \vec{e}(x,y,\omega_m) \times \vec{h}^*(x,y,\omega_m) \right] \cdot \hat{e}_z dxdy \right|^2}{\iint \left\{ \text{Re}\left[ \vec{e}(x,y,\omega_m) \times \vec{h}^*(x,y,\omega_m) \right] \cdot \hat{e}_z \right\}^2 dxdy} \tag{8}$$

$$\bar{n}_2(\omega_m) = \frac{\varepsilon_0}{\mu_0} \frac{\iint n_2(x,y,\omega_m) n^2(x,y,\omega_m) \left| \vec{e}(x,y,\omega_m) \right|^2 \left| \vec{e}^*(x,y,\omega_m) \right|^2 dxdy}{\iint \left\{ \text{Re}\left[ \vec{e}(x,y,\omega_m) \times \vec{h}^*(x,y,\omega_m) \right] \cdot \hat{e}_z \right\}^2 dxdy} \tag{9}$$

where $n(x,y,\omega_m)$ and $n_2(x,y,\omega_m)$ are the linear and nonlinear RIs of the material at position $(x,y)$ at the frequency $\omega_m$, respectively. $\vec{e}(x,y,\omega_m)$ and $\vec{h}(x,y,\omega_m)$ are the electric and magnetic field distributions on the waveguide transverse plane. $\hat{e}_z$ is the unit vector along the propagation direction. $\varepsilon_0$ and $\mu_0$ are the dielectric constant and the permeability constant, respectively. Finally, the conversion efficiency $\eta$ (in the unit of dB) is defined as the ratio of the output idler power to the output signal power, that is

$$\eta(\text{dB}) = -10 \lg \frac{\left| A_I(L) \right|^2}{\left| A_S(L) \right|^2} \tag{10}$$

where $L$ is the physical length of the waveguide.

We have measured the output FWM spectra of 21-mm-long SRN ridge waveguides with different waveguide dimensions under various coupled pump powers and extracted the conversion efficiencies (CEs), when the signal wavelength and the incident signal power were fixed at 1953 nm and 10 mW, respectively. For example, Figure 5a,b shows the measured FWM spectra of the waveguide with $h = 380$ nm and $w = 950$ nm under a coupled pump power of 61.5 mW, and $h = 640$ nm and $w = 1320$ under a coupled pump power of 52.4 mW, respectively. CEs of -53.2 dB and -51.1 dB can be extracted from the spectra. It should be noted that there was already some idler light generated in the incident fiber. We have normalized the output idler powers to that coming from the fiber and found little difference on the CEs between the normalizations before and after. Figure 5c,d shows the measured and normalized CEs with respect to the coupled pump power for the waveguides with $h = 380$ nm and $w = 950$ nm, and $h = 640$ nm and $w = 1320$ nm, respectively. By solving the equations from (4) to (10), we can fit the measured CEs versus the coupled pump power and the nonlinear parameter $\gamma$ can be extracted to be 2.79 $\text{W}^{-1}\,\text{m}^{-1}$ for the waveguide with $h = 380$ nm and $w = 950$ nm and 4.62 $\text{W}^{-1}\,\text{m}^{-1}$ for the waveguide with $h = 640$ nm and $w = 1320$, respectively. With the extracted $\gamma$ value, the nonlinear index $n_2$ of the SRN material can also be calculated by using Equation (7). Here, we assume the whole nonlinear effect was contributed by the SRN material since the silica surroundings have a nonlinear RI orders lower than that of the SRN material [45].

We have summarized the linear and nonlinear properties of the fabricated SRN ridge waveguides with different waveguide dimensions in Table 1. While there were variations for these measured or extracted values, the fabricated waveguides generally exhibited a linear loss of ~ 2 dB/cm and a nonlinear index of ~$1.13 \times 10^{-18}$ $\text{m}^2/\text{W}$. These values are indeed consistent with that from literatures where the SRN material and waveguide were measured at 1550 nm [30].

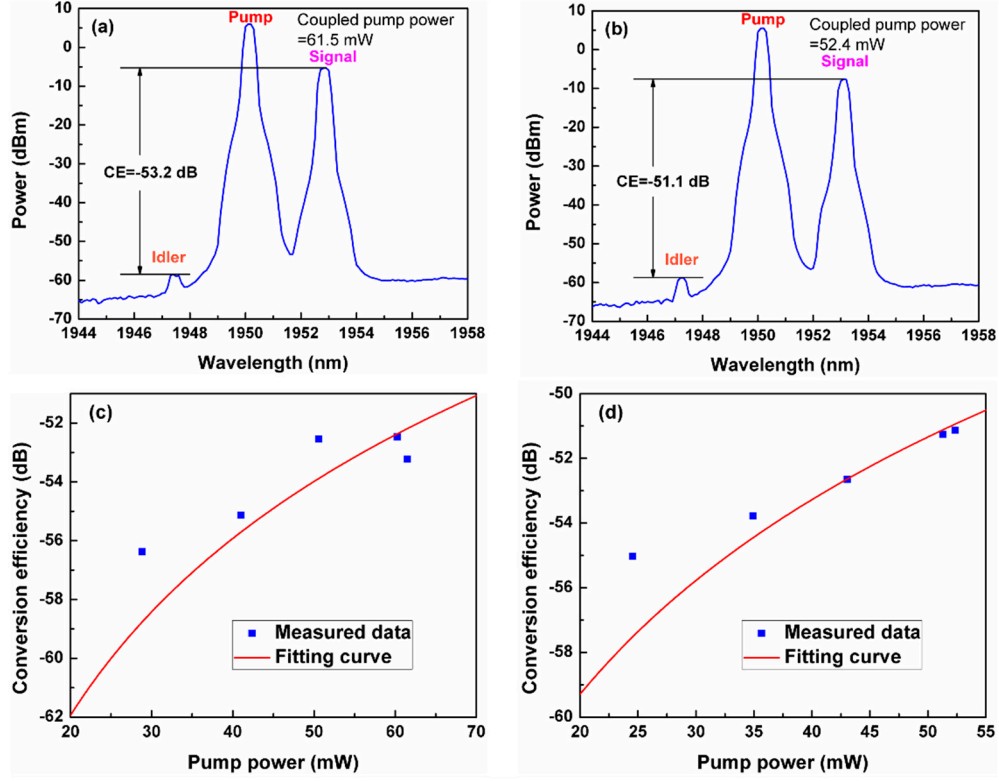

**Figure 5.** Measured output four-wave mixing spectra (**a,b**) and the wavelength conversion efficiencies with respect to the coupled pump powers (**c,d**) of the fabricated silicon-rich nitride ridge waveguides with a length of 21 mm. (**a**) and (**c**) are for the waveguide with $h$ = 380 nm and $w$ = 950 nm. (**b**) and (**d**) are for the waveguide with $h$ = 640 nm and $w$ = 1320 nm.

**Table 1.** Summary of the linear and nonlinear properties of the fabricated silicon-rich nitride ridge waveguides with various rib heights ($h$) and widths ($w$) at 1950 nm.

|  | Measured $\alpha$ (dB/cm) | Calculated $\alpha$ (dB/cm) | $\gamma$ (W$^{-1}$ m$^{-1}$) | $n_2$ (m$^2$/W) |
|---|---|---|---|---|
| $h$ = 230 nm, $w$ = 840 nm | 1.7 ± 1.6 | 1.8 | 3.74 | $1.77 \times 10^{-18}$ |
| $h$ = 230 nm, $w$ = 1360 nm | 2.1 ± 1.0 | 1.9 | 2.90 | $1.30 \times 10^{-18}$ |
| $h$ = 380 nm, $w$ = 950 nm | 1.7 ± 1.0 | 2.1 | 2.79 | $0.87 \times 10^{-18}$ |
| $h$ = 380 nm, $w$ = 1350 nm | 2.3 ± 1.0 | 2.1 | 2.49 | $0.80 \times 10^{-18}$ |
| $h$ = 640 nm, $w$ = 800 nm | 3.3 ± 1.4 | 3.3 | 5.98 | $1.24 \times 10^{-18}$ |
| $h$ = 640 nm, $w$ = 1320 nm | 2.7 ± 0.9 | 2.6 | 4.62 | $1.13 \times 10^{-18}$ |

## 4. Optimal Design for the Maximal Nonlinear Energy Efficiency

For waveguides without nonlinear losses, we can use a figure of merit (FOM) to evaluate the nonlinear energy efficiency of the waveguide, this is the ratio of the waveguide nonlinear parameter to the linear loss, expressed as [46]

$$\text{FOM} = \frac{\gamma}{\alpha} \tag{11}$$

For the proposed SRN ridge waveguide, both $\gamma$ and $\alpha$ are dependent on the width and rib height. While a deeper etching and a moderately smaller width can give a stronger confinement and hence a larger nonlinearity, this also leaves more light overlapping the rough sidewalls and therefore gives a larger linear propagation loss. Thus, the FOM can be used to fairly evaluate the overall nonlinear efficiency of the SRN ridge waveguide with different dimensions. We have calculated the nonlinear

parameter, linear loss and FOM of the SRN ridge waveguides for many combinations of $w$ and $h$ by knowing the linear and nonlinear RIs of SRN and the fabrication quality (i.e., $L_c$ and $\sigma$) and shown them in Figure 6a–c, respectively. As expectations, $\gamma$ and $\alpha$ generally have the similar varying trend as the waveguide width or etch depth changes. This finally yielded a maximal value of 0.0804 W$^{-1}$ of the FOM at the rib height $h$ = 700 nm and width $w$ = 1100 nm. Here, $\gamma$ an $\alpha$ were calculated to be 5.50 W$^{-1}$ m$^{-1}$ and 2.97 dB/cm, respectively.

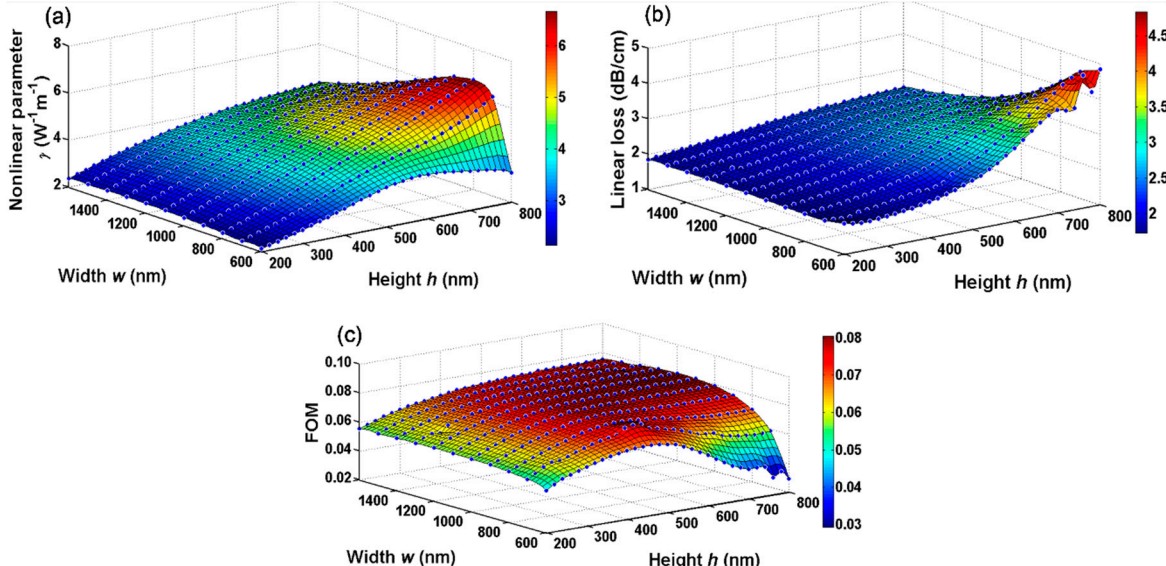

**Figure 6.** The calculated waveguide nonlinear parameter $\gamma$ (**a**), linear loss $\alpha$ (**b**) and figure of merit (FOM) (**c**) of a silicon-rich nitride ridge waveguide with respect to the width $w$ and rib height $h$. Here, the total height $H$ is fixed at 800 nm.

We have also calculated the nonlinear performances of the FWM wavelength conversion for the designed SRN ridge waveguide with the maximal FOM ($h$ = 700 nm, $w$ = 1100 nm). Figure 7a shows the CEs as the waveguide length increases, when the input pump wavelength/power and input signal wavelength/power were set to be 1950 nm/500 mW and 1951 nm/1 mW, respectively. Here, CE is defined as the ratio of the output idler power to the input signal power. The nonlinear effect for generating new idler photons dominated the power of the idler for a short waveguide while for a long waveguide, the linear loss took charge. Thus, there was an optimal length for maximal CE and, here in Figure 7a, a length of 14.6 mm was obtained for a maximal CE of –36.2 dB. It is worth noting that, to obtain a higher CE, one can further increase the pump power or use a resonating structure to enhance the nonlinear interactions between light and the SRN ridge waveguide. Figure 7b shows the calculated wavelength dependence of the CE for the optimized waveguide with a length of 14.6 mm at a fixed pump wavelength of 1950 nm and an adjusted signal wavelength. The 3 dB conversion bandwidth was found to be 94 nm.

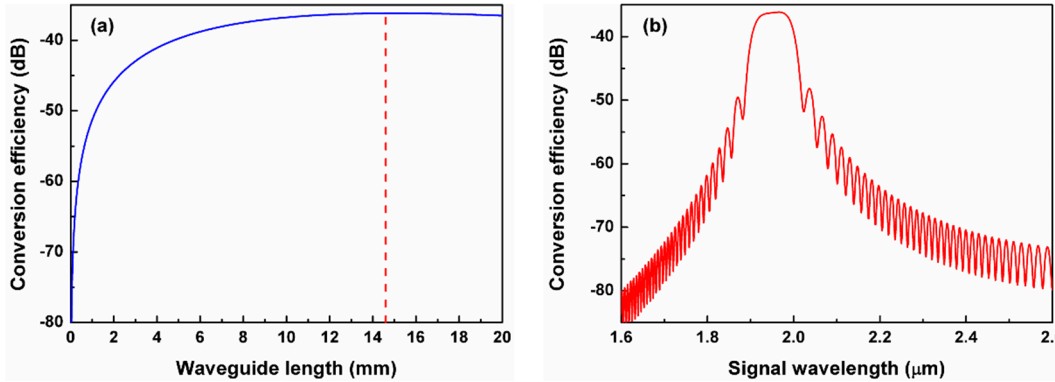

**Figure 7.** The calculated four-wave mixing conversion efficiency for the designed silicon-rich nitride ridge waveguide ($h$ = 700 nm, $w$ = 1100 nm) with a maximal figure of merit (FOM). (**a**) Dependence on the waveguide length. Here, difference of the pump and signal wavelengths is 1 nm. (**b**) Dependence on the signal wavelength. Here, the waveguide length is 14.6 mm and the pump wavelength is 1950 nm.

## 5. Optimal Design for a Superbroad FWM Conversion Bandwidth

Although the optimally designed waveguide above exhibits a maximal nonlinear efficiency, the bandwidth is still very limited due to a dispersion which is not small enough (164.3 ps/nm/km) at 1950 nm as shown by the calculated wavelength dependence of the dispersion in Figure 8a (green solid line). Figure 8a also shows the calculated wavelength-dependent dispersion curves for the SRN ridge waveguides with some other dimensions. The dispersion was found to be close to 0 (−6.0 ps/nm/km) at 1950 nm for the waveguide with $h$ = 625 nm and $w$ = 1050 nm. Such a low dispersion is expected to give a broad nonlinear bandwidth. For such an SRN ridge waveguide, the waveguide nonlinear parameter and the linear loss were calculated to be 5.24 $W^{-1}m^{-1}$ and 2.88 dB/cm, respectively, thus producing a FOM of 0.0790 $W^{-1}$. This FOM is only a little compromised compared with the maximal one (0.0804 $W^{-1}$). Furthermore, the calculated FWM CE was calculated to exhibit a maximal value of −36.22 dB for a waveguide length of 15 mm, when the input pump wavelength/power and input signal wavelength/power were set to be 1950 nm/500 mW and 1951 nm/1 mW, respectively. Figure 8b shows the calculated wavelength dependence of the FWM CE for the designed SRN ridge waveguide and the 3 dB conversion bandwidth was found to be as broad as 518 nm thanks to the small waveguide dispersion around 2 μm. Besides, one can also find that it may be not a good idea to etch the SRN layer through, i.e., $h$ = 800 nm, see the solid brown curve in Figure 8a, for a low dispersion in the 2 μm band, which is nevertheless mostly implemented in the 1550 nm band.

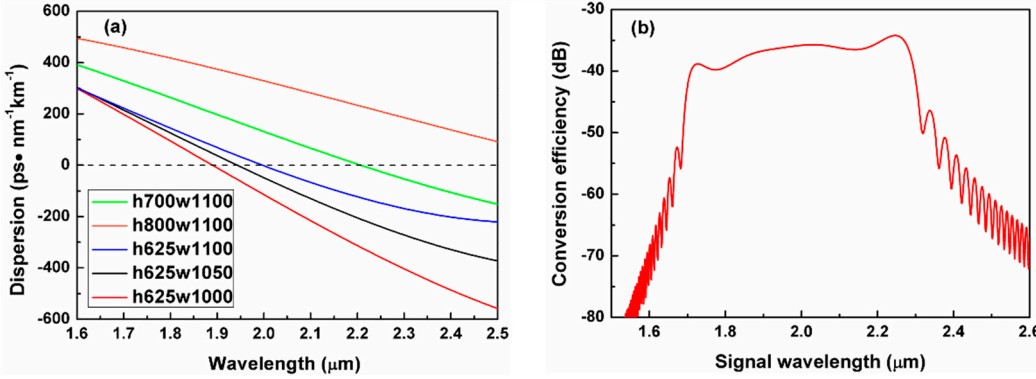

**Figure 8.** (**a**) Calculated wavelength dependence of the dispersion for the silicon-rich nitride ridge waveguides with various waveguide dimensions. (**b**) Calculated four-wave mixing conversion efficiency with respect to the signal wavelength for a SRN ridge waveguide ($h$ = 625 nm, $w$ = 1050 nm) with the zero-dispersion wavelength at 1950 nm.

## 6. Discussion and Conclusions

We have proposed and fabricated the silicon-rich nitride ridge waveguides and characterized their linear and nonlinear performances at 2 μm wavelengths. SRN ridge waveguides with different rib heights and widths were fabricated, exhibiting a linear loss around 2 dB/cm. Four-wave mixing experiments with a CW pump at 1950 nm were performed in the fabricated SRN ridge waveguides and revealed a waveguide nonlinear parameter of ~3-6 $W^{-1}m^{-1}$ around 2 μm. With the measured and extracted parameters which characterize the fabrication quality and the material property, optimal design of the SRN ridge waveguides was carried out and a maximal FOM of 0.0804 $W^{-1}$ was found among ridge waveguides with various dimension configurations at 1950 nm. Meanwhile, the ridge waveguide could also be designed to achieve the FWM conversion with a superbroad bandwidth (518 nm) and little compromise of the nonlinear FOM. First, these results show that a stripe silicon nitride waveguide may not be the best choice under some fabrication quality and material properties when moving the operating wavelengths from the conventional telecommunication band to longer wavelengths like the 2 μm band. Second, although the nonlinear conversion efficiency (–51.1 dB) achieved in our fabricated SRN ridge waveguide was much smaller than that of a silicon waveguide (–10 dB [24]) and an SRN waveguide (–18 dB [39]) pumped with pulses, it is indeed the first time, to the best of our knowledge, the nonlinear properties of a silicon nitride waveguide in the 2 μm spectral window have been revealed using a CW pump. Last, the proposed design method for optimal nonlinear performances, in terms of either FOM or the bandwidth, is expected to open new avenues for achieving better on-chip nonlinear waveguides for applications involving longer wavelengths like the 2 μm band.

**Author Contributions:** Conceptualization, design and fabrication, Z.T. and X.G.; Measurement, Z.T., and D.C.; Data analysis, Z.T., S.G. and X.G.; Writing—original draft preparation, Z.T.; writing—review and editing, D.C., H.H., S.G. and X.G.; All authors have read and agreed to the published version of the manuscript.

**Funding:** National Natural Science Foundation of China (Grant No. 61875172 and 61475138), Zhejiang Provincial Natural Science Foundation of China (Grant No. LD19F050001 and LY19F050014), Det Frie Forskningsråd (Danish Council for Independent Research) (DFF-7107-00242) and Villum Fonden (Villum Foundation) (023112, 00023316 and 15401).

**Conflicts of Interest:** The authors declare no conflict of interest.

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
