# Peer review of "Characterization and Optimal Design of Silicon-Rich Nitride Nonlinear Waveguides for 2 μm Wavelength Band"

_applsci, doi:10.3390/app10228087_

Round 1

Reviewer 1 Report

The authors propose and experimentally demonstrate the silicon-rich nitride ridge waveguides and characterize their linear and nonlinear performances at 2 μm wavelengths. I found the study comprehensive. The authors have fully demonstrated the concept in the experiment. The results are clearly presented and the manuscript is organized well. I recommend the publication of the work after addressing the following minor comments:

1- It is better that the authors discuss the implication of the study and the key problems that should be addressed in the future. Usually, the last part of the abstract should address this.

2- Fig. 1c has very low quality. It is very hard to see the values of the x and y axes.  

3- Is \omega in Eq. 1 the optical frequency or the radial optical frequency?

4- The conversion efficiency of the proposed ridge waveguide, reported in Fig. 7, is very small.. Could the authors comment on this?

Reviewer 2 Report

In this manuscript, Tu et al report the propagation loss and nonlinear performance of SRN waveguide at 2um. Optimal design is investigated and proposed with better FOM and wider bandwidth as compared to the experimental results. The novelty is justified in a solid way and the manuscript is well organized and written clearly. I would recommend the publication of this manuscript in Applied Sciences after minor revision:

  1. In Line 138 and 139, the authors mention Lc and σ2 can be fitted using Equation (2). More details should be provided. How is g obtained? What is the explicit relation between Lc and a2? What is the definition of d in Equation (3)?
  2. Continued from Question 1. σ is chosen to be 5 nm according to Ref43. But Ref 43 is Si waveguide. Please justify why this number can be directly used for SRN waveguide.
  3. Please explain the discrepancy between the measured data and fitting curve in Fig.5c and 5d.
  4. A comparison table comparing the nonlinear performance of this work ti the nonlinear performance of Si/SiN counterparts at 2um is desired.

Reviewer 3 Report

The authors deal with the realization of silicon  rich nitrite to study  the loss property.  

The paper is suitable  for publication  after the following issue.

1) in table 1 the error on the measured alpha are large. The authors should be highlight the motivations. 

2) the nonlinear terms  could be augmented  and what are the advantages to have triste term.
